# EMPIRICAL ANALYSIS OF THE HESSIAN OF OVER-PARAMETRIZED NEURAL NETWORKS

## ABSTRACT

We study the properties of common loss surfaces through their Hessian matrix. In particular, in the context of deep learning, we empirically show that the spectrum of the Hessian is composed of two parts: (1) the bulk centered near zero, (2) and outliers away from the bulk. We present numerical evidence and mathematical justifications to the following conjectures laid out by Sagun et al. (2016): Fixing data, increasing the number of parameters merely scales the bulk of the spectrum; fixing the dimension and changing the data (for instance adding more clusters or making the data less separable) only affects the outliers. We believe that our observations have striking implications for non-convex optimization in high dimensions. First, the *flatness* of such landscapes (which can be measured by the singularity of the Hessian) implies that classical notions of basins of attraction may be quite misleading. And that the discussion of wide/narrow basins may be in need of a new perspective around over-parametrization and redundancy that are able to create *large* connected components at the bottom of the landscape. Second, the dependence of a small number of large eigenvalues to the data distribution can be linked to the spectrum of the covariance matrix of gradients of model outputs. With this in mind, we may reevaluate the connections within the data-architecture-algorithm framework of a model, hoping that it would shed light on the geometry of high-dimensional and non-convex spaces in modern applications. In particular, we present a case that links the two observations: small and large batch gradient descent appear to converge to different basins of attraction but we show that they are in fact connected through their flat region and so belong to the same basin.

## 1 INTRODUCTION

In this paper, we study the geometry of the loss surface of supervised learning problems through the lens of their second order properties. To introduce the framework, suppose we are given data in the form of input-label pairs, $\mathcal{D} = \{(x^i, y^i)\}_{i=1}^N$ where $x \in \mathbb{R}^d$ and $y \in \mathbb{R}$ that are sampled i.i.d. from a possibly unknown distribution $\nu$, a model that is parametrized by $w \in \mathbb{R}^M$; so that the number of examples is $N$ and the number of parameters of the system is $M$. Suppose also that there is a predictor $f(w, x)$. The supervised learning process aims to solve for $w$ so that $f(w, x) \approx y$. To make the '$\approx$' precise, we use a non-negative loss function that measures how close the predictor is to the true label, $\ell(f(w, x), y)$. We wish to find a parameter $w^*$ such that $w^* = \arg\min \mathcal{L}(w)$ where,

$$\mathcal{L}(w) := \frac{1}{N} \sum_{i=1}^N \ell(f(w, x^i), y^i). \tag{1}$$

In particular, one is curious about the relationship between $\mathcal{L}(w)$ and $\hat{\mathcal{L}}(w) := \int \ell d(\nu)$. By the law of large numbers, at a given point $w$, $\mathcal{L}_w \to \hat{\mathcal{L}}_w$ almost surely as $N \to \infty$ for fixed $M$. However in modern applications, especially in deep learning, the number of parameters $M$ is comparable to the number of examples $N$ (*if not much larger*). And the behaviour of the two quantities may be drastically different (for a recent analysis on provable estimates see (Mei et al., 2016)).

A classical algorithm to find $w^*$ is gradient descent (GD), in which the optimization process is carried out using the gradient of $\mathcal{L}$. A new parameter is found iteratively by taking a step in the direction of the negative gradient whose size is scaled with a constant step size $\eta$ that is chosen from

line-search minimization. Two problems emerge: (1) Gradient computation can be expensive, (2) Line-search can be expensive. More involved algorithms, such as Newton-type methods, make use of second-order information (Nocedal & Wright, 2006). Under sufficient regularity conditions we may observe: $\mathcal{L}(w + \Delta w) \approx \mathcal{L}(w) + \Delta w \nabla \mathcal{L}(w) + \Delta w^T \nabla^2 \mathcal{L}(w) \Delta w$. A third problem emerges beyond an even more expansive computational cost of the Hessian: (3) Most methods require the Hessian to be non-degenerate to a certain extent.

When the gradients are computationally expensive, one can alternatively use its stochastic version (SGD) that replaces the above gradient with the gradient of averages of losses over *subsets* (such a subset will be called the *mini-batch*) of $\mathcal{D}$ (see (Bottou, 2010) for a classical reference). The benefit of SGD on real-life time limits is obvious, and GD may be impractical for practical purposes in many problems. In any case, the stochastic gradient can be seen as an approximation to the true gradient, and hence it is important to understand how the two directions are related to one another. Therefore, the discussion around the geometry of the loss surface can be enlightening in the comparison of the two algorithms: Does SGD locate solutions of a different nature than GD? Do they follow different paths? If so, which one is better in terms of generalization performance?

For the second problem of expensive line-search, there are two classical solutions: using a small, constant step size, or scheduling the step size according to a certain rule. In practice, in the context of deep learning, the values for both approaches are determined heuristically, by trial and error. More involved optimal step size choices involve some kind of second-order information that can be obtained from the Hessian of the loss function (Schaul et al., 2013). From a computational point of view, obtaining the Hessian is extremely expensive, however obtaining some of its largest and smallest eigenvalues and eigenvectors are not that expensive. Is it enough to know only those eigenvalues and eigenvectors that are large in magnitude? How do they change through training? Would such a method work in SGD as well as it would on GD?

For the third problem, let's look at the Hessian a little closer. A critical point is defined by $w$ such that $||\nabla \mathcal{L}(w)|| = 0$ and the nature of it can be determined by looking at the *signs* of its Hessian matrix. If all eigenvalues are positive the point is called a local minimum, if $r$ of them are negative and the rest are positive, then it is called a saddle point with index $r$. At the critical point, the eigenvectors indicate the directions in which the value of the function locally changes. Moreover, the changes are proportional to the corresponding -signed- eigenvalue. Under sufficient regularity conditions, it is rather straightforward to show that gradient-based methods converge to points where the gradient is zero. Recently Lee et al. (2016) showed that they indeed converge to minimizers. However, a significant and untested assumption to establish these convergence results is that the Hessian of the loss is non-degenerate. A relaxation of the above convergence to the case of non-isolated critical points can be found in (Panageas & Piliouras, 2016). What about the critical points of machine learning loss functions? Do they satisfy the non-degeneracy assumptions? If they don't, can we still apply the results of provable theorems to gain intuition?

## 1.1 A HISTORICAL OVERVIEW

One of the first instances of the comparison of GD and SGD in the context of neural networks dates back to the late eighties and early nineties. Bottou (1991) points out that large eigenvalues of the Hessian of the loss can create the illusion of the existence of local minima and GD can get stuck there, it further claims that the help of the inherent noise in SGD may help to get out of this obstacle. The origin of this observation is due to Bourrely (1989), as well as numerical justifications. However, given the computational limits of the time, these experiments relied on low-dimensional neural networks with few hidden units. The picture may be drastically different in higher dimensions. In fact, provable results in statistical physics tell us that, in certain real-valued non-convex functions, the local minima concentrate at an error level near that of the global minima. A theoretical review on this can be found in (Auffinger et al., 2013), while Sagun et al. (2014) and Ballard et al. (2017) provide an experimental simulation as well as a numerical study for neural networks. They notably find that high error local minima traps do not appear when the model is *over-parametrized*.

These concentration results can help explain why we find that the solutions attained by different optimizers like GD and SGD often have comparable training accuracies. However, while these methods find comparable solutions in terms of training error there is no guarantee they generalize equally. A recent work in this direction compares the generalization performance of *small batch* and

*large batch* methods (Keskar et al., 2016). They demonstrate that the *large batch* methods always generalize a little bit worse even when they have similar training accuracies. The paper further makes the observation that the basins found by *small batch* methods are wider, thereby contributing to the claim that wide basins, as opposed to narrow ones, generalize better.

The final part of the historical account is devoted to the observation of flatness of the landscape in neural networks and its consequences through the lens of the Hessian. In the early nineties, Hochreiter & Schmidhuber (1997) remarks that there are parts of the landscape in which the weights can be perturbed without significantly changing the loss value. Such regions at the bottom of the landscape are called *the flat minima*, which can be considered as another way of saying a *very wide minima*. It is further noted that such minima have better generalization properties and a new loss function that makes use of the Hessian of the loss function has been proposed that targets the *flat minima*. The computational complexity issues have been attempted to be resolved using the $R$-operator of Pearlmutter (1994). However, the new loss requires all the entries of the Hessian, and even with the $R$-operator, it is unimaginably slow for today's large networks. More recently, an *exact* numerical calculation of the Hessian has been carried out by Sagun et al. (2016). It turns out that the Hessian can have near zero eigenvalues even at a given random initial point, and that the spectrum of it is composed of two parts: (1) the bulk, and (2) the outliers. The bulk is mostly full of zero eigenvalues with a fast decaying tail, and the outliers are only a handful which appears to depend on the data. This implies that, locally, most directions in the weight space are flat, and leads to little or no change in the loss value, except for the directions of eigenvectors that correspond to the large eigenvalues of the Hessian.

## 1.2 OVERVIEW OF RESULTS

In this work, we present a phenomenological study in which we provide various observations on the local geometry at the bottom of the landscape and discuss their implications on certain features of solution spaces: Connectedness of basins found by large and small batch methods.

1. **Flatness at the bottom of the landscape:** At the bottom, most of the eigenvalues in the spectrum of the Hessian are near zero, except for a small number of relatively larger ones.

2. **A possible explanation through over-parametrization:** The decomposition of the Hessian as a sum of two matrices, where the first one is the sample covariance matrix of the gradients of model outputs and the second one is the Hessian of the function that describes the model outputs. We argue that the second term can be ignored as training progress which leaves us with the covariance term which leads to degeneracy in the Hessian when there are more parameters than samples.

3. **Dependence of eigenvalues to model-data-algorithm:** We empirically examine the spectrum to uncover intricate dependencies within the data-architecture-algorithm triangle: (1) more complex data produce more outliers, (2) increasing the network size doesn't affect the density of large eigenvalues, (3) large batch methods produce the same number of outliers that are larger in magnitude, and finally (4) there are negative eigenvalues even after the training process appears to show no further progress but their magnitude is much smaller than the outliers.

4. **A new interpretation of basins:** It has been a common practice to discuss certain features of basins found by various algorithms. Some recent examples, such as Keskar et al. (2016); Chaudhari et al. (2016); Jastrzębski et al. (2017), appear to draw the big-picture of isolated basins at the bottom of the landscape. One tool that is commonly used to demonstrate such claims is to evaluate the loss on a line that connects two solutions found by different methods. First of all, the notion of the basin itself can be misleading given the negative eigenvalues pointed out in the previous item. Moreover, we claim that this idea of isolated basins may be misleading based on the above observations of the dramatic level of flatness of the local geometry. In particular, we show that two solutions with different qualities can be shown to be in the same basin even with a loss evaluation on a straight line connecting the two.

**Remark 1:** Throughout this paper we use the notions of data complexity and over-parametrization vaguely. The complexity of data can be defined in various ways and further research is required

to determine the precise notion that is required that would link complexity to the spectrum. Over-parametrization, similarly, can be defined in various ways: $M >> N$, $M \to \infty$ for fixed $N$, or $M/N \to c$ for a certain constant, etc... However, more realistic notions of both complexity of the data and the over-parametrization should take the architecture into account, and detailed treatment of this should follow another work.

**Remark 2:** The notion of the basin, also, can be defined precisely. However, it is unclear whether any algorithm used in practice actually locates the bottom of a basin described in classical ways. For instance, the norm of the gradients are small but not at the machine precision, and the eigenvalues of the Hessian still has a negative part even after SGD continues a long while without a meaningful decrease in the loss value. This is presumably the fault of the algorithm itself, however, it requires a further study, and hence such notions of sharp vs. wide minima in various recent work should be taken with a grain of salt.

**Remark 3:** Even when one has a way to measure the width of a 'basin', such ways of measuring the approximate width are all relative. In a recent study, Dinh et al. (2017) shows how 'sharp minima' can still generalize with proper modifications to the loss function. We note that it takes a non-linear transformation to deform relative widths of basins. And, in this work, we focus on relative values as opposed to absolute values to get a consistent comparison across different setups.

## 2 SOME PROPERTIES OF THE HESSIAN

### 2.1 A FIRST LOOK AT THE SPECTRUM

We begin by an exact calculation of the spectrum of the Hessian at a random initial point, and at the end of training. Note that the plots are arranged in a way that they show the eigenvalues in the $y$-axis, and indicate the order of the eigenvalue in the $x$-axis. This choice is necessary to indicate the scale of the degeneracy while still showing all the eigenvalues in the same plot. Figure 1 shows the full spectrum of the Hessian at the random initial point of training and after the final training point. The model of this example is a two hidden layer network with a total of $5K$ parameters that is trained using gradient descent. Also note that, throughout the paper, the exact full Hessian is computed via the Hessian-vector products (Pearlmutter, 1994) up to the machine precision.

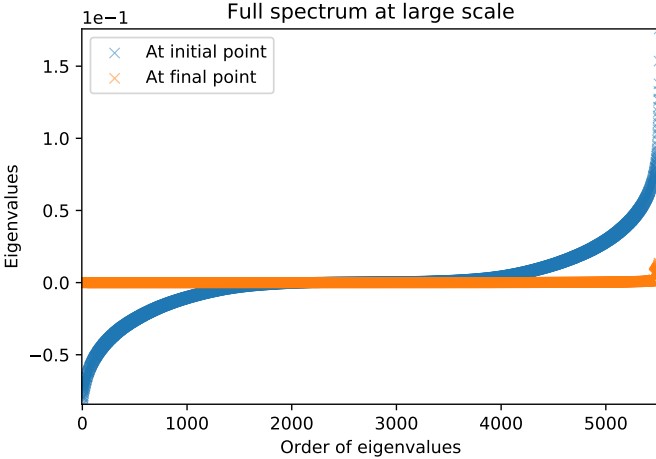

Figure 1: Ordered eigenvalues at a random initial point, and at the final point of GD. $x$-axis is the rank of the eigenvalue, and $y$-axis the value of the eigenvalue.

### 2.2 GENERALIZED GAUSS-NEWTON DECOMPOSITION OF THE HESSIAN

In order to study its spectrum, we will describe how the Hessian can be decomposed into two meaningful matrices (LeCun et al., 1998; Martens, 2010). Suppose the loss function is given as a composition of two functions, the model function $f_\bullet : \mathbb{R}^M \longrightarrow \mathbb{R}$ is the real-valued output of a network that depends on the parameters; and the loss function $\ell_\bullet : \mathbb{R} \longrightarrow \mathbb{R}^+$ is a convex function.

Here, $\bullet$ refers to the given example. Examples include the regression: the mean-square loss composed with a real-valued output function, and classification: the negative log-likelihood loss composed with the dot product of the output of a softmax layer with the label vector.

For ease of reading, we indicate the dependencies of functions $\ell$ and $f$ to data by the index of the example, or omit it altogether in case it is not necessary, and unless noted otherwise the gradients are taken with respect to $w$. The gradient and the Hessian of the loss for a given example are given by

$$\nabla \ell(f(w)) = \ell'(f(w))\nabla f(w) \tag{2}$$
$$\nabla^2 \ell(f(w)) = \ell''(f(w))\nabla f(w)\nabla f(w)^T + \ell'(f(w))\nabla^2 f(w) \tag{3}$$

where $\bullet^T$ denotes the transpose operation (here the gradient is a column-vector). Note that since $\ell$ is convex $\ell''(s) \geq 0$ and we can take its square root which allows us to rewrite the Hessian of the loss as follows:

$$\nabla^2 \mathcal{L}(w) = \frac{1}{N}\sum_{i=1}^{N}[\sqrt{\ell_i''(f_i(w))}\nabla f_i(w)][\sqrt{\ell_i''(f_i(w))}\nabla f_i(w)]^T + \frac{1}{N}\sum_{i=1}^{N}\ell_i'(f_i(w))\nabla^2 f_i(w) \tag{4}$$

In general, it isn't straightforward to discuss the spectrum of the sums of matrices by looking at the individual ones. Nevertheless, looking at the decomposition, we can still infer what we should expect. At a point close to a local minimum, the average gradient is close to zero. However, this doesn't necessarily imply that the gradients for individual samples are also zero. However, if $\ell'(f(\hat{w}))$ and $\nabla^2 f(\hat{w})$ are not correlated, then we can ignore the second term. And so the Hessian can be approximated by the first term:

$$\nabla^2 \mathcal{L}(\hat{w}) \approx \frac{1}{N}\sum_{i=1}^{N}[\sqrt{\ell_i''(f_i(\hat{w}))}\nabla f_i(\hat{w})][\sqrt{\ell_i''(f_i(\hat{w}))}\nabla f_i(\hat{w})]^T \tag{5}$$

Here, Equation 5 is the sum of rank one matrices (via the outer products of gradients of $f$ multiplied by some non-negative number), therefore, the sum can be written as a product of an $M \times N$ matrix with its transpose where the columns of the matrix are formed by the scaled gradients of $f$. Immediately, this implies that there are at least $M - N$ many trivial eigenvalues of the right-hand side in Equation 5.

From a theoretical point of view, the tool that is required for the above problem should be a mapping of eigenvalues of the population matrix to the sample covariance matrix. Recent provable results on this can be found in Bloemendal et al. (2016) (please refer to the appendix for a review of this approach). We emphasize that this result require independent inputs, and extensions to correlated data appear to be unavailable to the best of our knowledge.

## 3  IMPLICATIONS OF FLATNESS FOR THE GEOMETRY OF THE ENERGY LANDSCAPE

In this section, we leave the decomposition behind and focus on experimental results of the spectrum of the full Hessian through the exact Hessian-vector products. We discuss how data, model, and algorithm affect the spectrum of the Hessian of the loss.

### 3.1  THE RELATION BETWEEN DATA AND EIGENVALUES

In many cases of practical interest, the data contains redundancies. In such cases, the number of non-trivial eigenvalues could be even smaller than $N$. For instance, if one deals with a classification problem where the training data has $k$ classes with relatively small deviation among each of them, it is reasonable to expect that there will be an order of $k$ many non-trivial eigenvalues of the first term of the above decomposition of the Hessian in Equation 5. Then, if the second term is small (for instance when all the gradients per example are zero), we would expect to see $k$ many outliers in the spectrum of the Hessian of the loss. To test this idea, we used a feed-forward neural network with a 100-dimensional input layer, two hidden layers each of which with 30 hidden units, and a $k$ dimensional

output layer that is combined with softmax for $k$-class classification. We randomly sampled from $k$ Gaussian clusters in the input space and normalized the data globally. Then we carried out the training using SGD on ReLU network for the following number of clusters: $k : \{2, 5, 10, 20, 50\}$.

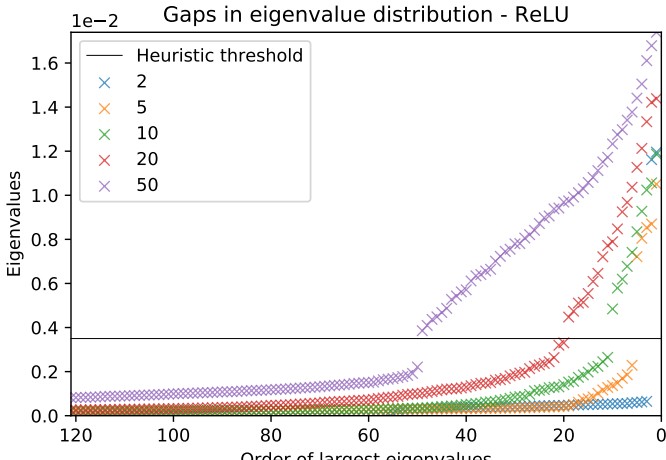

Figure 2: Ordered plot of eigenvalues with a visible gap. $x$-axis is the rank of the eigenvalue, and $y$-axis the value of the eigenvalue.

The number of large eigenvalues that are above the gap in Figure 2 match exactly the number of classes in the dataset. This experiment is repeated for various different setups. Please refer to Table 1 in the appendix for more experiments on this.

## 3.2 THE RELATION BETWEEN NUMBER OF PARAMETERS AND EIGENVALUES

We test the effect of growing the size of the network when data, architecture, and algorithm are fixed. In some sense, we make the system more and more over-parametrized. Based on the intutions developed above we should not observe a change in the size and number of the large eigenvalues of the Hessian at the bottom. To test this, we sample 1K examples from the MNIST dataset and fix them as the training set. Then we form four different networks each of which has a different number of nodes in its hidden layer ($n_{\text{hidden}} \in \{10, 30, 50, 70\}$). All four networks are trained with the same step size and the same number of iterations and the exact Hessian is computed at the end. Figure 3 shows the largest 120 eigenvalues of each of the four Hessians ranked in an increasing order. For the right edge of the spectrum (that is, for the large positive eigenvalues), the shape of the plot remains invariant as the number of parameters increase (Figure 3).

## 3.3 THE RELATION BETWEEN THE ALGORITHM AND EIGENVALUES

Finally, we turn to describing the nature of solutions found by the large and small batch methods for the training landscape. We train a convnet that is composed of 2 convolution layers with relu-maxpool followed by two fully-connected layers. The training set is a subsampled MNIST with 1K training examples. The small batch method uses a mini-batch size of 10, and the large-batch one uses 512. A learning rate for which both algorithms converge is fixed for both LB and SB. Note that the stochastic gradients are averaged over the mini-batch. Therefore, fixing the learning rate allows the algorithms to take steps whose lengths are proportional to the norms of the corresponding stochastic gradients averaged over the mini-batch. This way we ensure that both methods are compared fairly when we look at them after a fixed number of iterations. We train until the same number of iterations have been reached. Then, we calculate the Hessian and the spectrum of the Hessian in Figure 4. The large batch method locates points with larger positive eigenvalues.

This observation is consistent with Keskar et al. (2016) as the way they measure flatness takes local rates of increase in a neighborhood of the solution into account which is intimately linked to the size of the large eigenvalues.

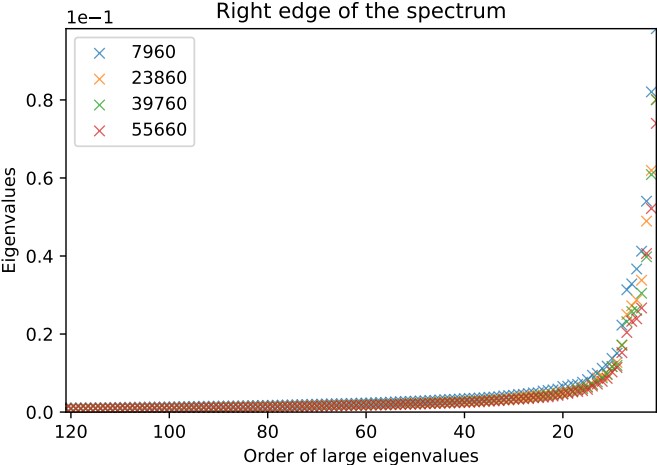

Figure 3: Right edge of the spectrum of the Hessian of the loss for a fully connected network on MNIST with increasing dimensions. $x$-axis is the rank of the eigenvalue, and $y$-axis the value of the eigenvalue.

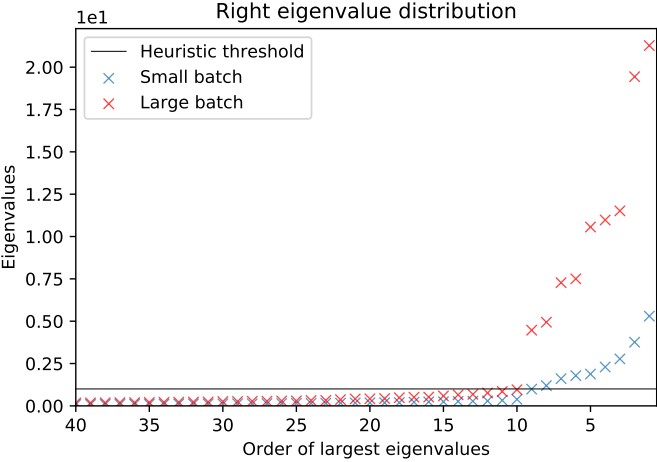

Figure 4: Outlier eigenvalues of LB ($bs = 512$) vs SB ($bs = 10$). $x$-axis is the rank of the eigenvalue, and $y$-axis the value of the eigenvalue.

### 3.4    TRAILING NEGATIVE EIGENVALUES AFTER TRAINING

Lastly, we observe that the negative eigenvalues at the end of the training are orders of magnitude smaller than the large ones. The very existence of the negative eigenvalues indicates that the algorithm didn't locate a local minimum, yet. Note that the stopping criterion in most practical cases is arbitrary. Training is stopped after there is no meaningful decrease in the loss value or increase in the test accuracy. In our experiments, the training ran well beyond the point of this saturation. In this time-scale, the loss decays in much smaller values. And we may expect convergence to a local-minimum at large (possibly exponentially long) time-scales. However, from a practical point of view, it appears that the properties of the landscape at this fine-grained scale is less relevant in terms of its test performance. Anyhow, we observe that there are a number of negative eigenvalues but their magnitude is much smaller compared to the positive eigenvalues.

The reason we look at the ranked negative eigenvalues in percentages rather than the order is the result of the experiment in Section 3.2. Adding more weights to the system scale the small scale eigenvalues proportionally, the *number* of outliers remain unchanged whereas the *ratio* of negative

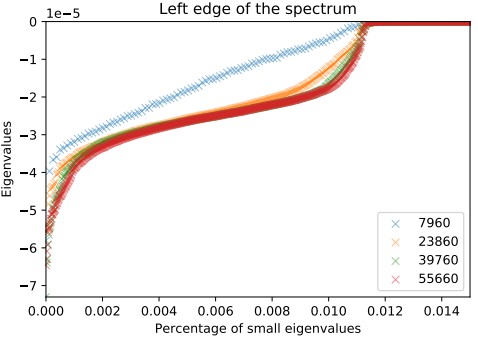 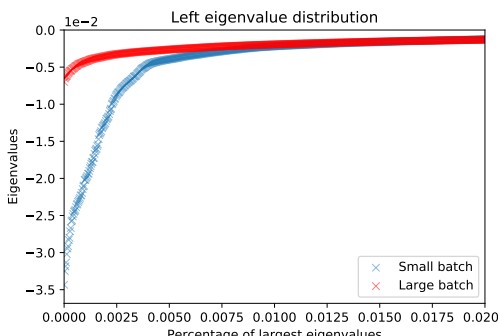

Figure 5: Negative eigenvalues at the bottom when increasing the number of hidden nodes (compare with Figure 3).

Figure 6: Negative eigenvalues at the bottom for LB-SB experiment on MNIST (compare with Figure 4).

$x$-axis indicates the order of the eigenvalue in percentages, $y$-axis indicates the eigenvalues.

eigenvalues remain the same. Moreover, they appear to be converging to have the same shape (Figure 6). Also, note that the negative eigenvalues can only come from the second term of the decomposition in Equation 5. Unless the second term contributes to the spectrum in an asymmetrical way, the observation that the negative eigenvalues are small confirms our previous suspicions that the effect of the ignored term in the decomposition is small.

## 4 DISCUSSION ON BASINS OF SOLUTIONS AND GENERALIZATION

Finally, we revisit the issue through the lens of the following question: What does over-parametrization imply on the discussion around GD vs. SGD (or large batch vs small batch) especially for their generalization properties? In this final section, we will argue that, contrary to what is believed in Keskar et al. (2016) and Chaudhari et al. (2016) the two algorithms do *not* have to be falling into different basins.

As noted in the introduction, for a while the common sense explanation on why SGD works well (in fact better) than GD (or large batch methods) was that the non-convex landscape had local minima at high energies which would trap large-batch or full-batch methods. Something that SGD with small batch shouldn't suffer due to the inherent noise in the algorithm. However, there are various experiments that have been carried out in the past that show that, for reasonable large systems, this is not the case. For instance, Sagun et al. (2014) demonstrate that a two hidden layer fully connected network on MNIST can be trained by GD to reach at the same level of loss values as SGD [1]. In fact, when the step size is fixed to the same value for both of the algorithms, they reach the same loss value at the same number of iterations. The training accuracy for both algorithms are the same, and the gap between test accuracies diminish as the size of the network increase with GD falling ever so slightly behind. It is also shown in Keskar et al. (2016) that training accuracies for both large and small batch methods are comparably good. Furthermore, Zhang et al. (2016) demonstrates that training landscape is easy to optimize even when there is no clear notion of generalization. Such observations are consistent with our observations: over-parametrization (due to the architecture of the model) leads to flatness at the bottom of the landscape which is easy to optimize.

When we turn our attention to generalization, Keskar et al. (2016) note that LB methods find a basin that is different than the one found by SB methods, and they are characterized by how wide the basin is. As noted in Figure 4, indeed the large eigenvalues are larger in LB than in SB, but is it enough to

---

[1]Another important observation in this work that will be useful for our purposes is that constant rate GD (as well as SGD) do not stop as training progresses, the loss decreases ever so slightly. This means that even in the flat region, there is a small amount of signal coming from the total gradients that allow the algorithm to keep moving on the loss surface. We emphasize that this doesn't contradict with the theory, but this implies that the convergence may be achieved at time scales much larger than the ones we observe in practice.

justify that they are in different basins, especially given the fact that the number of flat directions are enormous.

## 4.1 ARE THEY REALLY DIFFERENT BASINS? CASE OF CIFAR10 WITH LB VS SB

The observation that LB converges to sharper basins that are separated by wells from the wider basins found by SB has been an observation that triggered attention. In this section, we present two solutions with different qualities as measured by the generalization error and we show that they are in fact in the same 'basin' by showing that the evaluation of the loss doesn't go through a barrier between the two solutions. We start by two common pitfalls that one may fall into in testing this:

**The problem with epoch based time scales:** A common way to plot training profiles in larger scale neural networks is to stop every epoch to reserve extra computational power to calculate various statistics of the model at its current position. This becomes problematic when one compares training with different batch sizes, primarily because the larger batch model takes fewer steps in a given epoch. Recall that the overall loss is averaged, therefore, for a fixed point in the weight space, the empirical average of the gradients is an unbiased estimator for the expected gradient. Hence it is reasonable to expect that the norms of the large batch methods match to the ones of the small batch. And for a fair comparison, one should use the same learning rate for both training procedures. This suggests that a better comparison between GD and SGD (or LB and SB) should be scaled with the number of steps, so that, on average both algorithms are able to take similar number of steps of comparable sizes. The experiments we present use the number of iterations as the time-scale.

**The problem with line interpolations in the weight space:** The architecture of typical neural networks have many internal symmetries, one of which is the flip symmetry: when one swaps two nodes (along with the weights connected to it) at a given layer, the resulting network is identical to the original one. Therefore, when one trains two systems to compare, it may well be possible that the two fall into different flip symmetrical configurations that may look more similar when they are reordered. Therefore, training two systems with levels of randomness (seed, batch-size, choice of the initial point, etc.) may result in two points in the weight space that present a barrier only because of such symmetries. In an attempt to partially alleviate this problem we switch dynamics of an already trained system.

1. *Part I:* Train full CIFAR10 data for a bare AlexNet (bare meaning: no momentum, no dropout, and no batch normalization) with a batch-size of $1,000$. Record every 100 steps for 250 times.

2. *Part II:* Continue training from the endpoint of the previous step with a smaller batch-size of 32. Everything else, including the constant learning rate is kept the same. And train another 250 periods each of which with 100 steps.

The key observation is the jump in the training and test losses, and a drop in the corresponding accuracies (Figure 8). Toward the end of *Part II* the small batch reaches to a slightly better accuracy (about $\sim 1\%$). And this looks in line with the observations in Keskar et al. (2016), in that, it appears that the LB solution and SB solutions are separated by a barrier and that the latter of which generalizes better. Moreover, the line interpolations extending away from either endpoint appear to be confirming the sharpness of LB solution. However, we find the *straight line* interpolation connecting the endpoints of *Part I* and *Part II* turns out to *not* contain any barriers (Figure 9). This suggests that while the *Part I* and *Part II* converge to two solutions with different properties, these solutions have been in the same basin all along. This raises the striking possibility that those other seemingly different solutions may be similarly connected by a flat region to form a larger basin (modulo internal symmetries).

Another interpretation of this experiment, also, goes through the Gauss-Newton decomposition introduced in Equation 5. When we decrease the batch size, we increase the noise in the covariance of the gradients, and hence the first term starts to dominate. Even when the weight space has large flat regions, the fluctuations of the stochastic noise should be precisely in the directions of the large eigenvalues. Therefore, at the beginning of Part II, the loss increases because the point fluctuates in the directions corresponding to the large eigenvalues, and eventually settles at a point that lies at the interior of the same level set, essentially staying in the same basin.

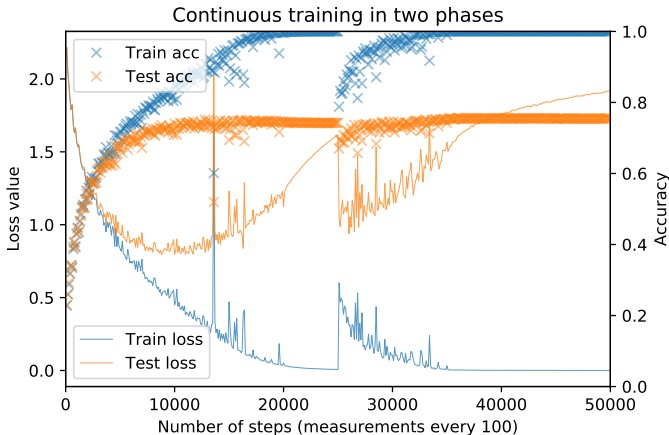

Figure 8: Large batch training immediately followed by small batch training on the full dataset of CIFAR10 with a *raw* version of AlexNet. The accuracy increases by about $1\%$: *Part I* and *Part II* locate solutions with different generalization properties.

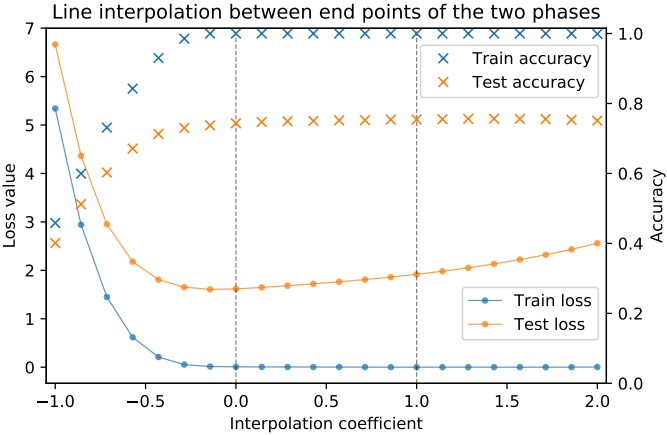

Figure 9: Loss and accuracy evaluation on the *straight* line that contains LB and SB solutions. The accuracy of the SB solution is $\sim 1\%$ better than the LB solution, but there is no barrier between the two points.

## 4.2 FURTHER DISCUSSIONS

One of the most striking implications of flatness may be the connected structure of the solution space. We may wonder whether two given solutions can be connected by a continuous path of solutions. This question has been explored in a recent work: in Freeman & Bruna (2016) it is shown that for one hidden layer rectified neural networks the solution space is connected which is consistent with the flatness of the landscape. The classical notion of basins of attractions may not be the suitable objects to study for neural networks. Rather, we may look at the exploration of interiors of level sets of the landscape. We may be tempted to speculate that such an exploration may indeed result in point that generalizes better. However, the flat space itself is very high dimensional which comes with its own computational issues.

The training curve can be seen as composed of two parts: (1) high gain part where the norm of the gradients are large, (2) noise of the gradients is larger relative to the size of the stochastic gradients (see (Shwartz-Ziv & Tishby, 2017) for a recent reference). We speculate that the first part is relatively easy and even a large batch method can locate a large level set that contains points that generalize better than what's initially found. From a practical point of view, using larger batches with larger step sizes can, in fact, accelerate training. An example of this can be found in Goyal et al. (2017),

where training Imagenet with a minibatch size of 8192 can match small batch performance. On a final note for further consideration, we remark that we used standard pre-processing and initialization methods that are commonly used in practice. Fixing these two aspects, we modified the data, model, and algorithm in order to study their relative effects. However, the effects of pre-processing and initialization on the Hessian is highly non-trivial and deserves a separate attention.

## 5 CONCLUSION

We have shown that the level of the singularity of the Hessian cannot be ignored from theoretical considerations. Furthermore, we use the generalized Gauss-Newton decomposition of the Hessian to argue the cluster of zero eigenvalues are to be expected in practical applications. This allows us to reconsider the division between initial fast decay and final slow progress of training. We see that even large batch methods are able to get to the same basin where small batch methods go. As opposed to the common intuition, the observed generalization gap between the two is not due to small batch finding a different, better, wider basin. Instead, the two solutions appear to be in the same basin. This lack of a barrier between solutions is demonstrated by finding paths between the two points that lie in the same level set. To conclude, we propose a major shift in perspective on considerations of the energy landscape in deep learning problems.

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

## A OUTLIER EIGENVALUES

In the subsequent experiments, we used a feed-forward neural network with a 100 dimensional input layer, two hidden layers each of which with 30 hidden units, and a $k$ dimensional output layer that is combined with softmax for $k$-class classification. We sampled random $k$ Gaussian clusters in the input space and normalized the data globally. Then we carried out the training for the following sets of parameters: $k : \{2, 5, 10, 20, 50\}$, algorithm: {GD, SGD}, non-linearity: {tanh, ReLU}, initial multipler for the covariance of the input distribution: $\{1, 10\}$. Then we counted the number of large eigenvalues according to three different cutoff methods: largest consecutive gap, largest consecutive ratio, and a heuristic method of determining the threshold by searching for the elbow in the scree plot (see Figure 2). In Table 1 we marked the ones that are off by $\pm 1$.

| alg-size | cov | mean&std | nc | gap | ratio | heur | cutoff |
|---|---|---|---|---|---|---|---|
| GD tanh 4022 | 1.0 | 1.19e-05, 1.19e-05 | 2 | **2** | **3** | N/A | N/A |
| GD tanh 4115 | 1.0 | 2.52e-05, 2.52e-05 | 5 | 2 | 17 | N/A | N/A |
| GD tanh 4270 | 1.0 | 3.75e-05, 3.75e-05 | 10 | 2 | 4 | N/A | N/A |
| GD tanh 4580 | 1.0 | 5.37e-05, 5.37e-05 | 20 | 3 | 4 | N/A | N/A |
| GD ReLU 4022 | 1.0 | 7.13e-06, 7.13e-06 | 2 | **3** | 4 | **2** | 0.003 |
| GD ReLU 4115 | 1.0 | 1.55e-05, 1.55e-05 | 5 | **6** | 7 | **5** | 0.003 |
| GD ReLU 4270 | 1.0 | 3.04e-05, 3.04e-05 | 10 | **11** | 12 | **10** | 0.003 |
| GD ReLU 4580 | 1.0 | 5.65e-05, 5.65e-05 | 20 | 4 | **21** | **21** | 0.003 |
| GD ReLU 5510 | 1.0 | 1.16e-04, 1.16e-04 | 50 | **50** | 51 | 49 | 0.003 |
| SGD ReLU 4022 | 1.0 | 8.96e-06, 8.96e-06 | 2 | **3** | 4 | **2** | 0.002 |
| SGD ReLU 4115 | 1.0 | 1.62e-05, 1.62e-05 | 5 | **6** | 7 | 7 | 0.002 |
| SGD ReLU 4270 | 1.0 | 2.97e-05, 2.97e-05 | 10 | 2 | 14 | 15 | 0.002 |
| SGD ReLU 4580 | 1.0 | 4.53e-05, 4.53e-05 | 20 | 2 | 3 | **21** | 0.002 |
| SGD ReLU 5510 | 1.0 | 6.78e-05, 6.78e-05 | 50 | **50** | 51 | 49 | 0.002 |
| SGD ReLU 4022 | 10.0 | 9.49e-05, 9.49e-05 | 2 | **2** | 4 | **2** | 0.015 |
| SGD ReLU 4115 | 10.0 | 1.68e-04, 1.68e-04 | 5 | **5** | **6** | **5** | 0.015 |
| SGD ReLU 4270 | 10.0 | 1.92e-04, 1.92e-04 | 10 | 2 | **11** | 9 | 0.015 |
| SGD ReLU 4580 | 10.0 | 3.10e-04, 3.10e-04 | 20 | **20** | **21** | 19 | 0.015 |
| SGD ReLU 5510 | 10.0 | 1.67e-04, 1.67e-04 | 50 | **50** | 51 | 49 | 0.005 |

Table 1: Counting outliers for matching the number of blobs. Dictionary of table elements: {cov: scale of covariance for inputs, mean&std: of the eigenvalues, nc: number of clusters, gap: largest consecutive gaps, ratio: largest consecutive ratios, heur: heuristic threshold, cutoff: the value of the heur.}

## B THE SPECTRUM OF THE GENERALIZED GAUSS-NEWTON MATRIX

In this section, we will show that the spectrum of the Generalized Gauss-Newton matrix can be characterized theoretically under some conditions. Suppose that we can express the scaled gradient $(\sqrt{\ell_i''(f_i(\hat{w}))} \nabla f_i(\hat{w})$ from Equation 5) as $g = Tx$ with the matrix $T \in M \times d$ depending only on the parameters $w$ - which is the case for linear models. Then we can write: $G = \frac{1}{N} \sum_{i \in \mathcal{D}} g_i g_i^T = \frac{1}{N} T X X^T T^T$, where $X = \{x^1, \ldots, x^N\}$ is an $d \times N$ matrix. Furthermore, without loss of generality, we assume that the examples are normalized such that the entries of $X$ are independent with zero mean and unit variance. One of the first steps in studying $G$ goes through understanding its principle components. In particular, we would like to understand how the eigenvalues and eigenvectors of $G$ are related to the ones of $\Sigma$ where $\Sigma := \mathbb{E}(G) = \frac{1}{N} T X X^T T^T = T T^T$.

In the simplest case, we have $\Sigma = Id$ so that the gradients are uncorrelated and the eigenvalues of $G$ are distributed according to the Marčenko-Pastur law in the limit where $N, M \to \infty$ and $\alpha := \frac{M}{N}$. The result dates back to sixties and can be found in (Marčenko & Pastur, 1967). Note that if $M > N$ then there are $M - N$ trivial eigenvalues of $G$ at zero. Also, the width of the nontrivial distribution essentially depends on the ratio $\alpha$. Clearly, setting the expected covariance to identity is very limiting. One of the earliest relaxations appear in (Baik et al., 2005). They prove a phase transition for the largest eigenvalues of the sample covariance matrix which has been known as the *BBP phase transition*. A case that may be useful for our setup is as follows:

**Theorem 1** (Baik, Arous, Péché, et al., 2005). *If $\Sigma = diag(\ell, 1, \ldots, 1)$, $\ell > 1$, and $M, N \to \infty$ with $\frac{M}{N} = \alpha \geq 1$. Let $c = 1 + \sqrt{\alpha}$, and let's call the top eigenvalue of the sample covariance matrix as $\lambda_{max}$ then:*

- *If $1 \leq \ell < c$ then $\lambda_{max}$ is at the right edge of the spectrum with Tracy-Widom fluctuations.*

- *If $c < \ell$ then $\lambda_{max}$ is an outlier that is away from bulk centered at $\ell(1 + \frac{\alpha}{\ell-1})$ with Gaussian fluctuations.*

Typically, due to the correlations in the problem we don't have $\Sigma$ to be the identity matrix or a diagonal matrix with spikes. This makes the analysis of their spectrum a lot more difficult. A solution for this slightly more general case with non-trivial correlations has been provided only recently by Bloemendal et al. (2016). We will briefly review these results here see how they are related to the first term of the above decomposition.

**Theorem 2** (Bloemendal, Knowles, Yau, and Yin, 2016). *If $d = M$, $\Sigma - Id$ has bounded rank, $\log N$ is comparable to $\log M$, and entries of $X$ are independent with mean zero and variance one, then the spectrum of $\Sigma$ can be precisely mapped to the one of $G$ as $M, N \to \infty$ for fixed $\alpha = \frac{M}{N}$. Let $K = min\{M, N\}$, and the decomposition of the spectrum can be described as follows:*

- ***Zeros:*** *$M - K$ many eigenvalues located at zero (if $M > N$).*

- ***Bulk:*** *Order $K$ many eigenvalues are distributed according to Marčenko-Pastur law.*

- ***Right outliers:*** *All eigenvalues of $\Sigma$ that exceed a certain value produce large-positive outlier eigenvalues to the right of the spectrum of $G$.*

- ***Left outliers:*** *All eigenvalues of $\Sigma$ that are close to zero produce small outlier eigenvalues between 0 and the left edge of the bulk of $G$.*

*Moreover, the eigenvectors of outliers of $G$ are close to the corresponding ones of $\Sigma$.*

This theorem essentially describes the way in which one obtains outlier eigenvalues in the sample covariance matrix assuming the population covariance is known. Here is an example:

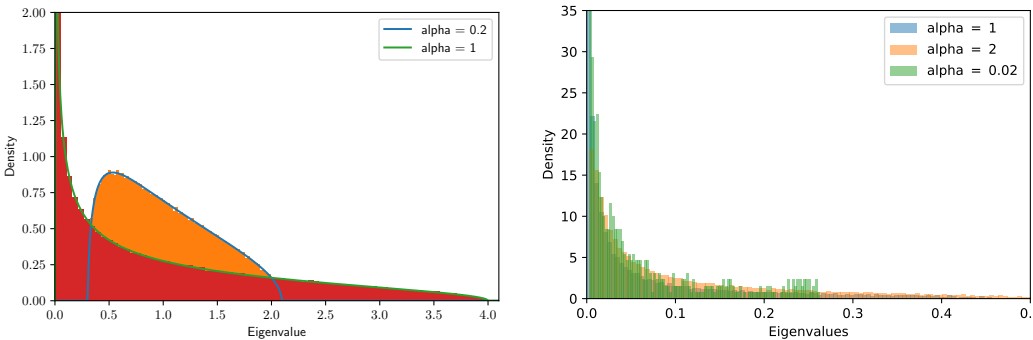

Figure 10: Spectrum of the logistic regression loss with tanh unit: when data has a single Gaussian blob (left), when data has two Gaussian blobs (right). In the latter case, the spectrum has outlier eigenvalues at 454.4, 819.5, and 92.7 for $alpha = 1, 2, 0.02$, respectively.

**Example 1** (Logistic regression). Consider the log-loss $\ell(s, y) = -y \log \frac{1}{1+e^{-s}} - (1 - y) \log(1 - \frac{1}{1+e^{-s}})$ and a single neuron with the sigmoid non-linearity. Note that $\ell(s, y)$ is convex in $s$ for fixed $y$, and we can apply the decomposition using $\ell$ and $f(w, x) = \langle w, x \rangle$. In this case we have $M = d$, also, note that the second part of the Hessian in Equation 4 is zero since $\nabla^2 f(w, x) = 0$. So the Hessian of the loss is just the first term. It is straightforward to calculate that the gradient per sample is of the form $g = c(w, x) Id_M x$ for a positive constant $c = c(w, x)$ that doesn't depend on $y$. This case falls into the classical Marčenko-Pastur law (left pane of Figure 10).

**Example 2.** Once we have more than one class this picture fails to hold. For, $\ell(s) = -\log(s)$, and $f(w; (x, y)) = \frac{\exp^{-\langle w_y, x \rangle}}{\sum_k \exp^{-\langle w_{y^k}, x \rangle}}$ the spectrum changes. It turns out that in that case the weights have one large outlier eigenvalue, and a bulk that's close to zero (right pane of Figure 10).

