# OpenReview forum: "Empirical Analysis of the Hessian of Over-Parametrized Neural Networks"
_ICLR.cc/2018/Conference — Invite to Workshop Track_

### Official Review · AnonReviewer1 · 2017-11-17
**A worthwhile set of experiments, but not entirely convincing, and confusingly presented.**

**Rating:** 5
**Confidence:** 2

**Review:**

The authors perform a set of experiments in which they inspect the Hessian matrix of the loss of a neural network, and observe that most of the eigenvalues are very close to zero. This is a potentially important observation, and the experiments were well worth performing, but I don't find them fully convincing (partly because I was confused by the presentation).

They perform four sets of experiments:

1) In section 3.1, they show on simulated data that for data drawn from k clusters, there are roughly k significant eigenvalues in the Hessian of the solution.

2) In section 3.2, they show on MNIST that the solution contains few large eigenvalues, and also that there are negative eigenvalues.

3) In section 3.3, they show (again on MNIST) that at their respective solutions, large batch and small batch methods find solutions with similar numbers of large eigenvalues, but that for the large batch method the magnitudes are larger.

4) In section 4.1, they train (on CIFAR10) using a large batch method, and then transition to a small batch method, and argue that the second solution appears to be better than the first, but that they are a part of the same basin (since linearly while interpolating between them they don't run into any barriers).

I'm not fully convinced by the second and third experiments, partly because I didn't fully understand the plots (more on this below), but also because it isn't clear to me what we should expect from the spectrum of a Hessian, so I don't know whether the observed specra have fewer large eigenvalues, or more large eigenvalues, then would be "natural". In other words, there isn't a *baseline*.

For the fourth experiment, it's unsurprising that the small batch method winds up in a different location in the same basin as the large batch method, since it was initialized to the large batch method's solution (and it doesn't appear to me, in figure 9, that the small batch solution is significantly different).

Section 2.1 is said to contain an argument that the second term of equation 5 can be ignored, but only says that if \ell' and \nabla^2 of f are uncorrelated, then it can be ignored. I don't see any reason that these two quantities should be correlated, but this is not an argument that they are uncorrelated. Also, it isn't clear to me where this approximation was used--everywhere? In section 3.2, it sounds as if the exact Hessian is used, and at the end of this section the authors say that figure 6 demonstrates that the effect of this second term is small, but I don't see why this is, and it isn't explained.

My main complaint is that I had a great deal of difficulty interpreting the plots: it often wasn't clear to me what exactly was being plotted, and most of the language describing them was frustratingly vague. For example, figure 6 is captioned "left edge of the spectrum, eigenvalues are scaled by their ratio". The text explains that "left edge of the spectrum" means "small but negative eigenvalues" (this would be better in the caption), but what are the ratios? Ratio of what to what? I think it would greatly enhance clarity if every plot caption described exactly, and unambiguously, what quantities were plotted on the horizontal and vertical axes.

Some minor notes:

There are a number of places where "it's" is used, where it should be "its".

In the introduction, the definition of \mathcal{L}' is slightly confusing, since it's an expectation, but the use of "'" makes one expect a derivative. Perhaps use \hat{\mathcal{L}} for the empirical loss, and \mathcal{L} for the expected one?

On the bottom of page 4, "if \ell' and \nabla f are not correlated": I think the \nabla should be \nabla^2.

It's "principal components", not "principle components".

---

> ### Author Response · Authors · 2018-01-05
> **Response to specific points**
>
> Thank you very much for your time to review our paper. We have added a joint response that should cover most of the issues addressed, and we updated the pdf file for our work. Here are some specific comments:
>
> The phenomena of a large number of eigenvalues being small (e.g. for Figure 1, 95% of the eigenvalues for the final point are within the band of [-10^(-4), -10^(-4)]) is a geometrical feature of the landscape that may change our way of visualizing the landscape. For instance, if one is to consider a random polynomial of degree 3 or more, and in a large number of variables, at a local minimum, the histogram of the eigenvalues of the loss function will be a shifted semi-circle distribution which is drastically different. Or in another context, if one is interested in the sample covariance function as in a perfect solution that ignores the second term, and if M < N and the data are iid then the spectrum would have a Marcenko Pastur part (see Appendix for more details). However, what we observe here doesn't fit into such provable cases, and to the best of our knowledge, there is no mathematically sound theoretical argument that would provide us with an explanation for the case at hand. Therefore, the scope of our work is to stick to the experiments and gain insight into what may actually be happening at the bottom of the loss landscape.
>
> Thank you also for pointing out the improvements, we have edited the text to reflect on increasing the clarity of our experiments, and exposition. We hope that our message is better conveyed this way.

---

### Official Review · AnonReviewer2 · 2017-11-28
**The assumption for the numerical analysis is not sound and more experiments need to be conducted**

**Rating:** 4
**Confidence:** 4

**Review:**

This paper studies the spectrum of the Hessian matrix for neural networks. To explain the observation that the spectrum of Hessian is composed of a bulk of eigenvalues centered near zero and several outliers away from the bulk, it applies the generalized Gauss-Newton decomposition on the Hessian matrix and argues that the Hessian can be approximated by the average of N rank-1 matrices. It also studies the effects on the spectrum from the model size, input data distribution and the algorithm empirically. Finally, this paper revisits the issue that if SGD solutions with different batch sizes converge to the same basin.

Pros:
1. The spectra of the Hessians with different model sizes, input data distributions and algorithms are empirically studied, which provides some insights into the behavior of over-parameterized neural networks.
2. A decomposition of the Hessian is introduced to explain the degeneracy of the Hessian. Although no mathematical justification for the key approximation Eq. (6) is provided, the experiments in Sec. 3 and Sec. 4 seem to suggest the analysis and support the approximation.

Cons:
1. The paper's contributions seem to be marginal. Many arguments in the paper have been first brought out in Sagun et. al.(2016) and Keskar et. al.(2016): the degeneracy of the Hessian, the bulk and outlier decomposition of the Hessian matrix and the flatness of loss surface at basins. The authors failed to show the significance of their results. For example, what further insights do the results in Sec. 3 provide to the community compared with Sagun et. al.(2016) and Keskar et. al.(2016).

2. More mathematical justification is needed. For example, in the derivation of Eq (6), why can we assume l'(f) and the gradient of f to be uncorrelated? How does this lead to the vanishing of the second term in the decomposition?

3.  More experiments are needed to support the arguments. For example, Sec. 4.1 shows that the solutions of SB SGD and LB SGD fall into the same basin, which is opposed to the results of Keskar et. al. (2016). However, this conclusion is not convincing. First, this result is drawn from one dataset. Second, the solution of SB SGD is initialized from the solution of LB SGD. As claimed in Keskar et. al. (2016), the solution of LB SGD may already get trapped at some bad minimum and it is not certain if SB SGD can escape from that. If it can't, then SB and LB can still be in the same basin as per the setting in this paper. So I'd like to suggest the author compare SB and LB when random initializations are conducted for both algorithms.

4.  In general, this paper is easy to read. However, it is not well organized. In the introduction, the authors spent several paragraphs for line search and expensive computation of GD and the Hessian, which I don't think are very related to the main purpose of this paper. Besides, the connection between the analysis and the experimental results is very weak and should be better established.

Minor points:
1. Language is awkward in section 3.1 and 3.2: 'contains contains', 'more smaller than', 'very close zero'...
2. More experimental details need to be included, such as the parameters used in training and generating the synthetic dataset.
3. The author needs to provide an explanation for the disagreement between Figure (10) and the result of Keskar et. al.(2016). What's the key difference in experimental settings?

---

> ### Author Response · Authors · 2018-01-05
> **Brief response to the comments**
>
> Thank you very much for the comments. We have addressed some of the main concerns above in a general statement. Please consider that response in your re-evaluation, as well. We fixed the minor points and added more details to the experiments. We also changed the structure of the paper to emphasize our contribution and make it clearer.
>
> To be more precise, we have a simple perspective that can also be interpreted as a warning sign when one is interested in questions related to the geometry of the bottom of the landscape. We have insights derived from our simple experiments and a demonstration how common ways of visualizations can be misleading. As pointed out, we improved our presentation in this regard.
>
> Regarding point 3: we present two solutions that are qualitatively different and they show signs of being in different basins (sharp/narrow) but they are in the same basin. We also point out that the barriers between solutions can appear depending on internal symmetries of the system, and our experiment addresses this issue as well. Please refer to the general comment above and the updated text for further details.

---

> > ### Comment · AnonReviewer2 · 2018-01-12
> > **Reply**
> >
> > Thanks for attempting to address my concerns. But the responses to Point 2 and Point 3 are not still convincing to me. In particular, the soundness of the assumption for the mathematical justification is still not addressed and the experimental setting comparing SB against LB is not well designed. Considering the overall novelty and the contribution of this paper, I keep my rating.

---

### Official Review · AnonReviewer3 · 2017-12-04
**A thought provoking tentative claim but exposition needs a lot of work.**

**Rating:** 5
**Confidence:** 4

**Review:**

This paper has at its core an interesting, novel, tentative claim, backed up by simple experiments, that small batch gradient descent and large batch gradient descent may converge to points in the same basin of attraction, contrary to the discussion (but not the actual experimental results) of Keskar et al. (2016). In general, there is a pressing need for insight into the qualitative behavior of gradient-based optimization and this area is of immense interest to many machine learning practitioners. Unfortunately the interesting tentative insights are surrounded by many unsubstantiated and only tangentially related theoretical discussions. Overall the paper has the appearance of lacking a sharp focus. This is a shame since I found the core of the paper very interesting and thought provoking.

Major comments:

While the paper has some interesting tentative experimental insights, the relationship between theory and experiment is complicated. The theoretical claims are vague and wide ranging, and are not all individually well supported or even tested by the experiments. Rather than including lots of small potential insights which the authors have had about what may be going on during gradient-based optimization, I'd prefer to see a paper with much tighter focus with a small number of theoretical claims well supported by experiments (it's fine if the experiments are simplistic as here; that's still interesting).

A large amount of the paper hinges on being able to ignore the second term in (6), and this fact is referred to many times, but the theoretical and experimental justification for this claim is very thin.

The authors mention overparameterization repeatedly, and it's in the title, but they never define it. It also doesn't appear to take center stage in their experimental investigations (if it is in fact critical to the experiments then it should be made clearer how).

Throughout this paper there is not a clear distinction between eigenvalues being zero and eigenvalues being close to zero, or similarly between the Hessian being singular and ill-conditioned. This distinction is particularly important in the theoretical discussion.

It would be helpful to be clearer about the differences between this work and that presented in Sagun et al. (2016).

Minor comments:

The assumption that the target y is real is at odds with many regression problems and practically all classification. It might be worth generalizing the discussion to multidimensional targets.

It would be good to have some citations to support the claim that often "the number of parameters M is comparable to the number of examples N (if not much larger)". With 1-dimensional targets as considered here, that sounds like a recipe for extreme overfitting and poor generalization. Generically based on counting constraints and free parameters one would expect to be able to fit exactly any dataset of N output values using a model with M free parameters. (With P-dimensional targets the relevant comparison would be M vs N P rather than M vs N).

At the end of intro to section 1, "loss is non-degenerate" should be "Hessian of the loss is non-degenerate"? Also, didn't the paper cited assume at least one negative eigenvalue at any saddle point, rather than non-degeneracy?

In section 1.1, it would be helpful to explain the precise sense in which "overparameterized" is being used. Hopefully it is in the sense that there are more parameters than needed for good performance at the true global minimum (the additional parameters helping with the process of *finding* a good minimum rather than its existence) or in the sense that M -> infinity for N "equal to" infinity. If it is in the sense that M >> N then I'm not sure of the relevance to practical machine learning.

It would be helpful to use a log scale for the plot in Figure 1. The claim that the Hessian is ill-conditioned depends on the condition number, which is impossible to estimate from the plot.

The fact that "wide basins, as opposed to narrow ones, generalize better" is not a new claim of the Keskar et al. paper. I'd argue it's well-known and part of the classical explanation of why maximum likelihood methods overfit and Bayesian ones don't. See for example MacKay, Information Theory Inference and Learning Algorithms.

"It turns out that the Hessian is degenerate at any given point" makes it sound like the result is a theoretical one. As I understand it, the experimental investigation in Sagun et al. (2016) just shows that the Hessian may often be ill-conditioned. As above, more clarity is also needed about whether it is literally degenerate or just approximately so, in which case ill-conditioned is probably a more appropriate word. Ill-conditioned is also more appropriate than singular in "slightly singular but extremely so".

How much data was used for the simple experiments in Figure 1? Infinite data? What data was used?

It would be helpful to spell out the intuition in "Intuitively, this kind of singularity...".

I don't think the decomposition (5) is required to "explain why having more parameters than samples results in degenerate Hessian matrices". Generically one would expect that with 1-dimensional targets, N datapoints and N + Q parameters, there would be a Q-dimensional submanifold of parameter space on which the loss would be zero. Of course there would be a few conditions needed to make this into a precise statement, but no need for assuming the second term is negligible.

Is the conventional decomposition of the loss into l o f used for the generalized Gauss Newton that f is a function only of the input to the neural net and the model parameters, but not the target? I could be wrong, but that was always my interpretation.

It's not clear whether the phrase "bottom of the landscape" used several times in the paper refers to the neighborhood of local minima or of global minima.

What is the justification for assuming l'(f(w)) and grad f(w) are not correlated? That seems unlikely to be true in general! Also spell out why this implies the second term can be ignored. I'm a bit skeptical of the claim in general. It's easy to come up with counterexamples. For example take l to be the identity (say f has a relu applied to it to ensure everything is well formed).

"Immediately, this implies that there are at least M - N trivial eigenvalues of the Hessian". Make it clear that trivial here means approximately not exactly zero (in which case a good word would be "small"); this follows since the second term in (5) is only approximately zero. In fact it should be possible to prove there are M - N values which are exactly zero, but that doesn't follow from the argument presented. As above I'd argue this analysis is somewhat beside the point since N should be greater than M in practice to prevent severe overfitting.

In section 3.1, "trivial eigenvalues" should be "non-trivial eigenvalues".

What's the relevance of using PCA on the data in Figure 2 when it comes to analyzing training neural nets? Also, is there any reason 2 classes breaks the trend?

What size of data was used for the experiments to plot figure 2 and figure 3? Infinite?

It's not completely clear what the takeaway is from Figure 3. I presume this is supporting the point that the eigenvalues of the Hessian at convergence consist of a bulk and outliers. The could be stated explicitly. Is there any significance to the fact that the number of clusters is equal to the number of outliers? Is this supporting some broader claim of the paper?

Figure 4, 5, 6 would benefit from being log plots, and make the claim that the bulk has the same shape independent of data much stronger.

The x-axis in Figure 5 is not "ordered counts of eigenvalues" but "index of eigenvalues", and in Figure 6 is not "ratios of eigenvalues" but ratio of the index. In the caption for Figure 6, "scaled by their ratio" is not clear.

I don't follow why Figure 6 confirms that "the effects of the ignored term in the decomposition is small" for negative eigenvalues.

In section 3.3, when saying the variances of the steps are different but the means are similar, it may interesting to note that the variance is often the dominant term and much greater in magnitude than the mean when doing SGD (at least that's what I've experienced).

What's the meaning of "elbow at similar levels"? What's the significance?

In section 4 it is claimed that overparameterization is what "leads to flatness at the bottom of the landscape which is easy to optimize". The bulk-outlier view suggests that adding extra parameters may just add extra dimensions to the flat region, but why is optimizing 100 values in a flat 100-dimensional space easier than optimizing 10 values in a flat 10-dimensional space?

In section 4.1, "fair comparison" is misleading since it depends on perspective. If one cares about compute time then certainly measuring steps rather than epochs would not be a fair comparison!

What's the relevance of the fact that random initial points in high-dimensional spaces are almost always nearly orthogonal (N.B. the "nearly" should be added)? This seems to be assuming something about the mapping from initial point to basin of attraction.

What's the meaning of "extending away from either end points appear to be confirming the sharpness of [the] LB solution"? Is this shown somewhere?

It would be helpful to highlight the key difference to Keskar et al. (which I believe is initializing SB training from LB point rather than from scratch). I presume the claim is that Keskar et al. only found their "inverted camel hump" linear interpolation results due to the random initialization, and that this would also often be observed for, say, two random LB-from-scratch trainings (which may randomly fall into different basins of attraction). If this is the intended point then it would be good to make this explicit.

In "the first terms starts to dominate", to dominate what? The gradient, or the second term in (5)? If the latter, what is the relevance of this?

Why "even" in "Even when the weight space has large flat regions"?

In the last paragraph of section 4.1, it might be worth spelling out that (as I understand it) the idea is that the small batch method finds itself in a poor region to begin with, since the average loss over an SB-noise-sized neighborhood of the LB point is actually not very good, and so there is a non-zero gradient through flat space to a place where the average loss over an SB-noise-sized neighborhood is good.

In section 5, "we see that even large batch methods are able to get to the level where small batch methods go" seems strange. Isn't this of training set loss? Isn't the "level" people care about the test set loss?

In appendix A, the meaning of consecutive in "largest consecutive gap" and "largest consecutive ratio" was not clear to me.

Appendix B is only referred to in a footnote. What is its significance for the main theme of the paper? I'd suggest either making it more prominent or putting it in a separate paper.

---

> ### Author Response · Authors · 2018-01-05
> **Response to the points raised**
>
> Thank you very much for the helpful review, please note that some of the major themes are addressed in the general comment above.
>
> - In the decomposition, multi-target case can be covered by
> $\ell(s_y, y) = -s_y + \log\sum_{y'}\exp{s_{y'}}$. It is indeed the case that the output independent of the target would be a conventional way to go, to do that, we will expand the decomposition to cover vector-valued outputs, too.
>
> - The strict saddle property in Lee et. al. assumes isolated (therefore non-degenerate) critical point.
>
> - Log scale plot for Figure 1 doesn't produce a meaningful plot, however, it might be worthwhile to note that 95% of the eigenvalues for the final point are within the band of [-10^(-4), -10^(-4)]
>
> - For Figure 1, 2, and 3 a thousand samples are generated from Gaussian clusters. This point is also addressed in Section 3.1. Also, the takeaway of section 3.1 and 3.2 is the relation between the outliers and data (and not the size of the model).
>
> - By the bottom of the landscape, we mean loss values near zero (but not at zero). To be more precise, for a non-negative function, f, we mean an element from the set {w:f(w)<epsilon}. To the best of our knowledge, the values of the global minimum, and/or the local minima are unknown in the case of deep learning loss functions.
>
> - Regarding the correlation in the second term, that's right, a more plausible argument would be the perfect classifier that has zero gradients on each of the examples.
>
> - In most cases, M>N without 'severe' overfitting, for example, for CIFAR-10 N=50K and M is usually several million.
>
> - PCA was a way to assess the complexity of the data and show its relation to the eigenvalues. But we decided to remove it since a notion of complexity of the data in this context should take the architecture into account. We added a remark on this in the text, as well.
>
> - In our experience, relative values of the variance and the mean of the gradients in SGD depends on the phase of the training. We will look into this in more detail.
>
> - By a 'fair comparison', we mean a fair comparison of what algorithm finds what kind of solution, assuming one is interested in the behavior of the algorithm itself. Otherwise, the real-life computational challenges depend on the hardware, too. For instance, one could increase the batch-size up to the saturation of the GPU and not lose time on it. Therefore, scaling the time axis with the number of epochs can be misleading in a broader context.
>
> - If one is to select random points on the sphere, the selected points become more and more orthogonal as the dimension of the sphere increases. We have experiments that show that this orthogonality is preserved for the trained points, too, if one starts from orthogonal initial points. This is not surprising given the geometry of high dimensional spaces. But we can follow up on this in another work.
>
> - "In the last paragraph of section 4.1, it might be worth spelling out that (as I understand it) the idea is that the small batch method finds itself in a poor region to begin with, since the average loss over an SB-noise-sized neighborhood of the LB point is actually not very good, and so there is a non-zero gradient through flat space to a place where the average loss over an SB-noise-sized neighborhood is good." This is a great point, but we are curious about the following: Is it the size of the noise, or the shape of it? We believe this should be investigated further in a separate context.
>
> - "In section 5, "we see that even large batch methods are able to get to the level where small batch methods go" seems strange. Isn't this of training set loss? Isn't the "level" people care about the test set loss?" Right, we meant 'the same basin'.
>
> - By largest consecutive gap, we mean the largest element of the set of consecutive gaps of eigenvalues when they are ordered on the real line. And similarly with the largest consecutive ratio. They are just ways of finding a separator in the spectrum. Some which seem to work better than the others but such a separator should depend on the notion of the complexity of the dataset, as well.  Also, we added a note to explain the relevance of Appx B. The theorem there is a tool that maps the eigenvalues of the population matrix to the sample covariance matrix but it is only valid for independent data. We also provide an example where it can work and fail at the end of the appendix.

---

### Public Comment · (anonymous) · 2017-12-17
**Approximate Hessian always Positive Semi-definite.**

Interesting and much need line of research. I have one question regarding your experiments.

In equation (6), you approximate a Hessian as sum of rank-1 matrices of the form vv^T. But these kind of matrices formed are always Positive Semi-definite. Given that how did you estimate the Hessian's to have negative eigenvalues in Figures. 1, 5, 6, 8?

---

> ### Author Response · Authors · 2018-01-05
> **Exact vs approximate Hessian**
>
> Thank you very much for your question. The experiments are the exact Hessian calculation, therefore, it reflects the existing negative eigenvalues. Of course, they can only come from the second term of the decomposition. We modified the text to reflect this.

---

### Author Response · Authors · 2018-01-05
**A joint response to the common themes for the reviews**

We thank all three reviewers for their time to evaluate our work, here we craft a response that we believe should address some of the points commonly raised by all three reviewers. We have edited the paper to enhance the message we are trying to convey, and we hope it is more expressive in its new state.

The focus of our work: The landscape at the bottom is flatter than the picture depicted in many recent papers (some of which are other fellow ICLR submissions e.g. https://openreview.net/forum?id=rJma2bZCW). Therefore we should revise our notions of 'basin' in a way that will address this feature.

Our work is phenomenological, and it addresses the shortcomings of certain ways of picturing the landscape, and it calls for a change. To this end:
(1) We demonstrate the local geometry at the bottom of the landscape and its intricate relations with the data, model, and algorithm.
(2) Then we show how the space of solutions can be vastly connected if one avoids rather simple pitfalls.

General remarks:

- Our work is an enhancement over Sagun et. al. (2016) in the following ways: (1) We present more experiments of the spectrum of the Hessian in various different setups, as well as a possible explanation. Therefore we solidify the claims in a more robust way. (2) Based on the key insight from the previous part, we present an experiment where two qualitatively different solutions are connected, thereby challenging some of the recent work by pointing out the fact that certain ways of visualization techniques can be misleading. (3) Finally, to the best of our knowledge, Sagun et. al. (2016) hasn't been published anywhere besides the ArXiv. We believe that our contribution is the necessary addition that would build on top of that work. This can also be seen by the reviews of that work has got: https://openreview.net/forum?id=B186cP9gx

- Our experiments don't rely on the decomposition. The decomposition is a tool to analyze the results and make predictions to be tested experimentally. All experiments are standalone. We edited the text to better reflect this fact. We also added more details on the experimental procedures.

- Certain notions such as the data complexity and over-parametrization are vague since making them more precise would require the details of the architecture, as well. Our focus is on the flattened weight vector, therefore, for now, it would be enough to consider cases where M>>N. However, future work will take a more detailed look into this.

---

### Decision · Program_Chairs · 2018-01-29
**ICLR 2018 Conference Acceptance Decision**

**Decision:**

Invite to Workshop Track

**Comment:**

Pros:
+ Builds in important ways on the work of Sagun et al., 2016.

Cons:
- The reviewers were very concerned that the assumption in the paper that the second term of Equation (6) is negligible was insufficiently supported, and this concern remained after the discussion and the revision.
- The paper needs to be more precise in its language about the Hessian, particularly in distinguishing between ill conditioning and degeneracy.
- The reviewers did not find the experiment very convincing because it relied on initializing the small-batch optimization from the end point of the large-batch optimization.  Again, this concern remained following the discussion and revision.

The area chair agrees with the authors' comments in their OpenReview post of 08 Jan. 2018 "A remark on relative evaluation," and has discounted the reviewers' comments about the relative novelty of the work.  It is important not to penalize authors for submitting their papers to conferences with an open review process, especially when that process is still being refined.

However, even discounting the remarks about novelty, there are key issues in the paper that need to be addressed to strengthen it (the 3 "cons" above), so this paper does not quite meet the threshold for ICLR Conference acceptance.

However, because it raises really interesting questions and is likely to provoke useful discussions in the community, it might be a good workshop track paper.